# TERRARIUM: REVISITING THE BLACKBOARD FOR STUDYING MULTI-AGENT ATTACKS

## ABSTRACT

A multi-agent system (MAS) powered by large language models (LLMs) could automate tedious user tasks like meeting scheduling that require collaboration. LLMs enable more nuanced protocols accounting for unstructured private data and users' personal constraints. However, this design exposes these systems to new problems from misalignment to attacks by malicious parties that compromise agents or steal user data.

In this paper, we propose a TERRARIUM framework for fine-grained studies on safety, privacy, and security. We repurpose the *blackboard* design, an early approach in multi-agent systems, to create a modular and configurable playground to support multi-agent collaborative tasks using LLMs. We, then, identify key attacks vectors like misalignment, malicious agents, compromised communication, and poisoned data. We implement three scenarios and add four representative attacks, demonstrating the flexibility of our framework. TERRARIUM provides necessary tools to study and quickly iterate over new designs and can further advance our community's efforts towards trustworthy multi-agent systems.

## 1 INTRODUCTION

Agents capable of perceiving and acting in arbitrary environments, while interacting with one another, constitute a multi-agent system (MAS), which naturally gives rise to situations of collaboration (Wurman et al., 2008; Parker, 1998), negotiation (Roughgarden, 2005), and conflict (Tambe, 2011). Recently, the advent of agents based on large language models (LLMs), equipped with extended action spaces through tool use, has expanded their accessible environments and substantially increased the potential for agent interaction. This enables new real-world applications, from maintaining a meeting calendar to optimizing energy consumption, while allowing unstructured data and contexts by leveraging capabilities of LLMs. Considering these potential formulations of MAS, it is important to study these systems in a well-defined, self-sustaining testbed to better understand its capabilities and vulnerabilities.

Specifically, in the real world scenarios human interactions are exposed to incidents of adversarial behavior that undermines others for personal gain. This phenomena is not unique to humans, and LLM-driven agents face similar malicious interactions that undermine desired outcomes whether by humans or other LLM-driven agents. Thus, agents that are trusted with private information or hold powerful capabilities become potential targets of malicious behavior, by humans or other agents, which can degrade utility, leak sensitive information, or delay coordination.

In this paper we introduce TERRARIUM, a framework for *observing* and *studying* multi-agent interactions in an isolated, configurable environments that support a broad set of adversarial vectors. Motivated by OpenAI Gymnasium (Brockman et al., 2016) which standardized *training* reinforcement learning agents, we aim to provide a common way to analyze safety, security, and privacy in MASs across a spectrum of environments. In this paper, we focus on cooperative and general-sum formulations, more precisely, Distributed Constraint Optimization Problems (DCOPs). We target problems that inherently require multiple agents with private data (i.e., cases where no single, monolithic agent can solve the task simply by ingesting all data) so that coordination and communication become key objects of study.

We identify three key properties of MAS that enable effective collaboration and evaluation and also provide grounds for attacks: (1) a joint, ground-truth global objective to enable measurable impact,

(2) agents with private information and different capabilities, and (3) complex communication patterns and trajectories. We use these properties to guide and develop an attack framework that targets communication and coordination: attack dimensions that do not appear in single-agent settings.

However, to realize these properties and conduct attacks on any part of the MAS, we require a modular and configurable agent design and a centralized communication mechanism to inspect agent's interactions. For this, we revisit an early MAS design using a *blackboard* architecture Erman et al. (1980b). We further identify five key abstractions of our TERRARIUM framework: agents, environment, blackboards, tools, and the communication protocol. We implement the framework across multiple levels: enabling different problems, communication protocols, and providing a blackboard that accommodates persistence and modularity through Model Context Protocol (MCP).

Our evaluation shows, first, that MAS systems achieve solid utility, allowing LLMs to solve complex DCOP problems with sophisticated coordination; and second, that the TERRARIUM enables systematic study of key attack vectors: misalignment, data stealing, and denial-of-service. Our framework can be extended to study new setups, attacks and defenses, supporting further research towards trustworthy multi-agent systems.

## 2 BACKGROUND

An *agent* is typically defined as an autonomous entity that perceives its environment and acts upon it to achieve certain goals (Wooldridge, 2009). A system composed of multiple interacting agents is referred to as a *multi-agent system* (MAS). By coordinating their actions (whether cooperatively or competitively), agents in an MAS can solve problems that are beyond the capabilities of any individual agent or a centrally controlled system (Jennings et al., 1998). The MAS paradigm, which originated in distributed artificial intelligence, emphasizes decentralized decision-making and has led to the development of various frameworks for agent communication, coordination, and negotiation over the past decades.

MAS techniques have been applied in a wide range of domains. For example, cooperative MAS deployments include fleets of warehouse robots coordinating storage and fulfillment (Wurman et al., 2008) and multi-robot teams in disaster response (Parker, 1998). Competitive (adversarial) MAS arise when stakeholders have divergent objectives, e.g., defender–attacker resource allocation modeled as Stackelberg security games (Tambe, 2011), or congestion management with selfish agents in routing games (Roughgarden, 2005). Despite this breadth, many canonical MAS formalisms are computationally difficult to solve optimally even in realistic settings—for instance, planning in decentralized POMDPs is NEXP-complete even for finite horizons (Bernstein et al., 2002), and computing Nash equilibria in general games is PPAD-complete (Daskalakis et al., 2009). Amid this landscape, the *distributed constraint optimization* (DCOP) framework provides a cooperative alternative that preserves key MAS characteristics—decentralized control, local objectives, and communication-based coordination—while admitting scalable exact/approximate algorithms and structure-exploiting methods for many practical problems (Fioretto et al., 2018). Consequently, we focus on DCOPs as the backbone for our study.

**DCOP.** DCOPs involve a set of agents selecting local actions to maximize a global utility, typically formulated as the sum of local utility or constraint functions (Fioretto et al., 2018). Classical DCOP algorithms, such as complete search (e.g., ADOPT) (Modi et al., 2005), message passing on pseudo-trees (e.g., DPOP) (Petcu & Faltings, 2005), and local/approximate methods (e.g., Max-Sum, DSA) (Farinelli et al., 2008; Zhang et al., 2005), trade off optimality guarantees, message and memory complexity, and anytime behavior, and they perform best when problem structure and communication protocols are explicitly defined and utilities are numeric and stationary. Building on this formalism, we extend the framework with LLM-based agents. Unlike the classical DCOP setting—where utilities are fixed symbolic functions and messages follow engineered schemas—our agents communicate in free-form natural language, and local constraints/objectives can be specified textually rather than solely as hand-coded numeric functions. This preserves the DCOP backbone while enabling systematic *security* evaluations of LLM-specific vulnerabilities—e.g., prompt injection, communication poisoning.

**Agent communication protocols.** Interacting agents in a MAS are required to communicate to achieve their objectives by utilizing a communication protocol that structures the rules and polices of communication between agents. Without this, performance degradation, inefficient token usage, and capability loss can emerge. Recent advancements in communication protocols such as the Agent2Agent (A2A) protocol (Google, 2025) that assigns agent cards to specialized agents for more efficient collaboration initialization and allows agents to communicate with each other via messages, tools, and artifacts. Another established protocol is the Agent Communication Protocol (ACP) (IBM, 2025) which also enables agent-to-agent communication for low-latency communication. These protocols allow structured communication between agents and are vital to efficient and safe MAS. However, given that most communication protocols are developed by companies, we lack an open-source framework for testing and benchmarking communication protocols in a controllable and isolated environment which TERRARIUM satisfies.

**Agent platforms and benchmarks.** LLM agents introduce significant security challenges such as indirect prompt injection attacks (Greshake et al., 2023) and context hijacking (Bagdasarian et al., 2024). Debenedetti et al. (2024) proposed AgentDojo, which is a widely used dynamic environment to study agents' security and utility across domains such as Workspace, Travel, Slack, and Banking. Beyond single-agent benchmarks, Abdelnabi et al. (2024) proposed a simulation benchmark for multi-agent interactive negotiation in a non-zero-sum game setup to study cooperation and competition dynamics. Abdelnabi et al. (2025) studied security and privacy attacks in agent-to-agent communication where an AI assistant communicates with external parties to develop complex plans such as travel booking. Despite progress in this area, we lack a canonical platform that is easily extendable, configurable, and adaptable to study diverse multi-agent safety challenges, which is what we propose in our work.

**Multi-agent security.** Increased adoption of LLM-based MAS has increased concerns about their security and privacy risks. Existing research demonstrates that known vulnerabilities such as prompt injection (Greshake et al., 2023) and jailbreaking (Anil et al., 2024) manifest more severely in multi-agent settings, where compromising a single agent enables the attack to propagate to all others (Lee & Tiwari, 2024). Inspired by networking security challenges, recent work has also analyzed MAS protocols and introduced attack strategies including Agent-in-the-Middle (He et al., 2025) and control-flow hijacking (Triedman et al., 2025). MASs have also been used to conduct attacks, such as jailbreaks, on other LLMs (Rahman et al., 2025; Abdelnabi et al., 2025).

**Blackboards.** A *blackboard* is a shared, structured workspace where heterogeneous agents post partial results, hypotheses, constraints, and goals for others to observe, refine, or refute. Historically, blackboard systems such as HEARSAY-II coordinated independent "knowledge sources" via a central store and scheduler, integrating evidence across abstraction layers to resolve uncertainty (Erman et al., 1980a; Nii, 1986a;b). In our multi-agent setting, multiple LLM-driven agents can similarly communicate and coordinate by appending proposals, commitments, and exception notes to a common log rather than engaging in bespoke pairwise messaging. Contemporary realizations span tuple-space designs (Linda-style generative communication) (Gelernter, 1985), append-only event logs for decoupled producers/consumers (Kreps et al., 2011), CRDT-backed shared documents for eventual consistency (Shapiro et al., 2011), and vector-indexed memories that enable retrieval-augmented reads (Lewis et al., 2020).

## 3 SYSTEM PROPERTIES AND USE CASES

Multi-agent systems (MAS) increasingly sit in the loop of high-stakes, data-rich applications such as coordination, scheduling, energy, and logistics. They expose multiple attack surfaces at once: valuable *goals* and *private objectives* that an adversary may subvert, abundant *private data* such as user attributes, constraints, and locations, and *communication channels* such as messages, blackboards, and tools. LLM-driven agents amplify both capability and risk: instructions arrive in natural language, tools are invoked via text APIs, and behaviors can be steered or poisoned through subtle prompt-level manipulations. Our aim is to study MAS properties against adversaries in settings that are realistic yet evaluable.

## 3.1 PROBLEM SETUP

We seek MAS problem classes that (i) admit a broad spectrum of attacks (exfiltration, poisoning, spoofing, delay/DoS, collusion), (ii) provide a well-defined, quantitative ground truth for evaluating success and failure, and (iii) are implementable with LLM-based agents.

We therefore focus on following system properties:

**Agents.** Agents are instantiated by LLMs (optionally tool-augmented), which read natural instructions, exchange messages, and output decisions/actions. This preserves the *agentic interface* used in practice while giving us programmatic control over inputs, channels, and logs.

*Joint goal.* Agents participate in a cooperative task with a well-defined global objective. A shared goal makes adversarial impact *measurable*: subverting coordination (e.g., degraded utility, violated constraints) becomes a scalar signal rather than anecdote. It also enables oracle baselines (optimal or attack-free solutions) for evaluation, i.e., by measuring task utility value gap to oracle, violation counts, regret etc.

*Agentic data and capabilities.* Each agent possesses private data, constraints, roles, tools, and actuation. This heterogeneous private state creates a rich privacy and security surface that enables attacks such as exfiltration, impersonation, and capability escalation, while simultaneously exercising the system's access control and disclosure policies.

*Inter-agent communication.* Agents communicate over addressable channels—pairwise, group, and broadcast—and across modalities such as text, structured JSON, and images. Selective, partially private communication both opens realistic attack vectors including eavesdropping, spoofing, poisoning, man-in-the-middle, and Sybil coalitions, and enables corresponding defenses including authentication, redundancy, and audits. To support realistic experiments and reproducible analysis, the design provides per-message recipient control by the sender, optional encryption and authentication tags, and logged transcripts for ground-truth analysis and post-hoc forensics.

**Problem formalization.** We now introduce a formalization of our cooperative environments, modeled as DCOPs, see Appendix A for full notation glossary.

**Definition 1** (DCOPs)**.** An instance is a tuple

$$\mathcal{P} = \langle A,\ H,\ \eta,\ X,\ D,\ \sigma,\ C,\ \mu,\ o,\ \rho,\ F^{\star},\ \Pi \rangle.$$

**Objective.** Given a joint policy $\pi = (\pi_i)_{i \in A}$ and context $c \sim \mu$, for each agent $i$ draw a *joint* assignment over its owned variables $x_{X_i} \sim \pi_i(\cdot \mid o_i(c))$ independently across $i$ (or set $x_{X_i} = \pi_i(o_i(c))$ if deterministic), and assemble $x = (x_v)_{v \in X}$ from the collection $\{x_{X_i}\}_{i \in A}$. Define the cooperative objective

$$F^{\star}(x; c) = \sum_{i \in A} \sum_{\beta=1}^{k_i} f^{\star}_{i,\beta}(x_{S_{i,\beta}}; c),$$

and the optimization problem

$$\max_{\pi \in \prod_{i \in A} \Pi_i} \mathbb{E}_{c \sim \mu}\, \mathbb{E}_{x \sim \pi(\cdot \mid c)}\big[\, F^{\star}(x; c)\,\big],$$

where $x \sim \pi(\cdot \mid c)$ denotes the joint assignment induced by the agent-wise draws $\{x_{X_i}\}_{i \in A}$. Note that here the $o_i(c)$ is private to agent $i$ and $c$ is not jointly observed by any agents. The pair $(X, F^{\star})$ induces a bipartite factor graph $G = (V_{\text{var}} \cup V_{\text{fac}}, E)$ with $V_{\text{var}} = X$, $V_{\text{fac}} = F^{\star}$, and edge $(x, f)$ iff $x \in \text{scope}(f)$. Algorithms may pass messages on $G$; in practice, we realize it using blackboards.

## 4 SYSTEM DESIGN

We introduce a modular framework for evaluating the behavior, security vulnerabilities, and safety considerations of multi-agent systems composed of LLM-based agents. First, we enforce a well-defined, joint goal among agents to obtain a ground-truth objective function that allows reliable

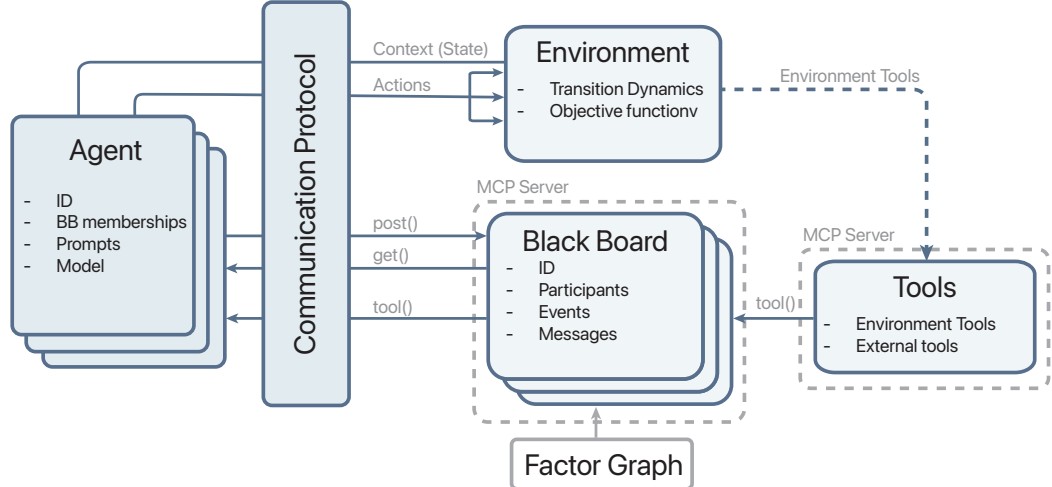

Figure 1: TERRARIUM is composed of multiple modules that are core to MAS communication and interaction such as agents, environment, blackboards, the communication protocol, and tools. For instance, an agent may post messages to a blackboard to communicate intentions and goals, call environment-dependent tools for search and observation (e.g., reading sensors for energy monitoring) or external tools (e.g., web search), and execute actions in the form of action tools that change the dynamics of the environment. The environment module is an isolated and self-sustaining simulator that takes in agent actions, synchronously or asynchronously, and gives each agent an observation (i.e., new context). To enable communication between agents, we initialize a set of blackboards through a configurable factor graph that determines the agent membership of each blackboard.

evaluation of joint agent policies in a controlled environment. Second, our framework enables diverse collections of agents, each with their own preferences, objectives, tools, and private data, mimicking that of agent characteristics of multi-agent systems in real-world scenarios. We consider the agent, environment, communication proxy, tools, and communication protocol as key modules composing TERRARIUM.

### 4.1 DESIRED PROPERTIES

Given the complexity of MAS, and the need for a simple, yet aligned framework to real-world implementations, there exist several desired properties that should be satisfied for effective experimentation, development, and evaluation. It is desirable to have a *modular* framework with extensive *configurability*. These properties enable more effective analysis on the capabilities, performance, and potential vulnerabilities of these systems.

**Modularity.** Multi-agent systems are often highly complex with its many interconnected components that can overwhelm researchers, engineers, and developers. Having an abstract framework that is simple and modular, allowing components to be swapped, will improve the usability and development of these systems. For example, this abstraction can enable effective ablations and focused experimentation on specific components for studies and analysis.

**Configurability.** In addition to modularity, we need full configurability of each module's parameters and allow different configurations for diverse MAS instances which is necessary for evaluating robustness of both performance and vulnerabilities. For example, this allows us to stress-test specific safety assumptions of a module and apply varying attacks to evaluate vulnerabilities. Additionally, this level of configurability improves reproducibility of MAS phenomena that is not as easily reproducible in environments where even one component is uncontrolled or unconfigurable.

## 4.2 DESIGN

We account for these desired characteristics in the design of TERRARIUM, where we decouple the *agents*, *environment*, *communication proxy*, *tools*, and *communication protocol* into distinct modules in favor of an abstract framework that aligns with current deployments of multi-agent systems. These properties allow fast and controlled experimentation for multi-agent analysis, security, and safety. Figure 1 outlines the proposed modularity and configurability.

A key module of our framework is the *communication proxy* that should enable fine-grained control and configurable observability of communication for which we use a set of Blackboards (Erman et al., 1980b). Each blackboard enables communication between two or more agents for coordination. The topology of the blackboards are instantiated by a factor graph, which determines the blackboard membership of agents, and implicitly controls the communication efficiency. For example, having all agents on one blackboard may not be efficient for communication since this could fill an agent's context of irrelevant conversations and information, resulting in degradation of performance even for agents with long context-lengths (Liu et al., 2023). We enable agents to have a predefined blackboard action set, enabling complex action sequences such as writing, erasing, referencing, reading, and highlighting. This may extend to a continuous action space with free-form actions rather than a discrete, token-level state space for more complex agent interaction behavior. Although, in our implementation, we adopt an append-only blackboard with reading and writing actions at a token-level for simplicity.

An *environment* consists of a function that receives actions from all agents, transitions its internal states, then sends the next state or context to the appropriate agent. Every environment has a well-defined, ground-truth joint objective function $F$ among agents which is used to evaluate action selection quality in a cooperative setting.

The *communication protocol* contains a set of rules and structures that organizes the communication between agents which is vital for efficient communication. We employ a simple communication protocol between agents that facilitates planning and action phases. Each agent are tasked to communicate intentions and goals to formulate a plan within a finite number of steps with access to blackboard tools (e.g., get(), post()), environment tools, and external tools to inform their decisions. Next, agents execute an action from a set of action tools that the environment provides and receive an updated context or state from the environment, detailing changes to the environment.

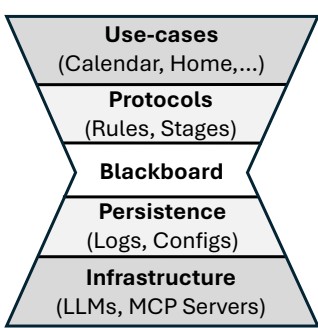

Figure 2: TERRARIUM implementation.

Finally, *agents* are modeled as an LLM that can take actions in the form of *tools* using model context protocol (MCP) (Hou et al., 2025) with FastMCP Lowin (2025). From here, we have full over the control the capabilities of the agents, their personalities, and internal objectives by using a layered stack as in Figure 2. Similar to how Wireshark (Combs, 1998) allows to study different networking protocols at different layers of the networking stack, TERRARIUM can be configured to support different backbone servers, configuration formats, communication protocols, and use-cases while connecting them through the blackboard primitives outlined in Figure 1.

## 5 ATTACKS

The interaction between agents during collaboration requires extensive information sharing to provide sufficient context for solving tasks. These conversations are facilitated by protocols specifically designed for multi-agent communication. However, the method, volume, and nature of information exchange introduce new challenges related to security, privacy, and system robustness.

The modular and flexible design of TERRARIUM enables systematic exploration of these challenges. Its configurability allows us to easily construct attack scenarios, while its modularity supports the creation of adversarial agents with diverse capabilities across different environments. Furthermore, its observability enables implementing and measuring Attack Success Rate (ASR) metrics.

## 5.1 ATTACK-ENABLING CHARACTERISTICS

A successful multi-agent design requires system and collaboration strategies with characteristics unique to a MAS, which adversaries can exploit to construct a range of attacks. Such attacks, leveraging these and other MAS-specific features, may have catastrophic effects on the system as the compromise propagates through multiple interactions.

***Actions can serve any objective*** In a collaborative environment, agents typically assume that the actions of others are intended to optimize the global objective. However, verifying that an agent's actions are not instead serving an ulterior objective might not be feasible. This creates unique challenges, as a large number of potential ulterior objectives can be pursued through actions that still appear valid. The ulterior motive can range from targeting a specific goal or agent to reducing the performance of the entire system.

***Agents needs external tools.*** To take actions and retrieve the information necessary for those actions, agents must rely on external tools. These tools can be leveraged for both active attacks, where the adversary modifies the interaction to carry out the attack, and passive attacks, where the adversary observes the interaction to infer information. While multi-agent poisoning has been studied in limited contexts, such as poisoning via web-retrieved content (Lee & Tiwari, 2024), the growing scale and complexity of MAS systems—and their increasing reliance on heterogeneous tool sets—introduce novel challenges.

***New MASs lead to new protocols.*** MASs are facilitated through protocols. These underlying protocols connect agents to one another and, in many cases, to central platforms that coordinate their interactions (Dang et al., 2025; Bhatt et al., 2025). As these protocols rapidly evolve and form the core of modern MASs, it becomes essential to study the novel attack vectors that emerge from their integration.

## 5.2 EXPLORING ATTACK VECTORS

The characteristics specified in the previous section motivate us to explore novel attack vectors to test the Confidentiality, Integrity and Availability (CIA) of MASs. We use these attack vectors to understand the security and privacy implications of MASs and to motivate future research.

***Confidentiality. Can agents keep a secret?*** To evaluate confidentiality, we examine whether agents preserve private information communicated by a previous agent when interacting with a subsequent one. In this setup, the second agent is instructed to elicit the personal information shared earlier, while the queried agent receives a system prompt explicitly stating that it should not reveal private information about any other agent. Our findings show that the *agent nevertheless discloses the information, despite being prompted not to*.

***Integrity. What actions lead to misalignment?*** To evaluate integrity, we examine misalignment by determining which actions, and the minimum number of actions, must be altered by an adversary during collaboration. We investigate two scenarios: an adversarial agent part of the collaboration and an external adversary capable of poisoning communication. The adversarial agent is restricted to modifying messages on its blackboard and communicating its own preferences, whereas the external adversary can target any agent, action, or blackboard. Our results indicate that (1) *a single adversarial agent is sufficient to misalign the entire system*, and (2) *an external adversary can induce misalignment in one shot with access to a single planning round, though multiple shots are required to improve efficacy*.

***Availability. How easily does context overflow?*** In collaborative settings, agents expend significant tokens and maintain extensive context windows to retain the history of prior interactions, which informs future decisions. We were able to carry out the availability attacks at only a fraction of

the cost required for completing the underlying task. These findings suggest that *MASs are particularly vulnerable to context overflow attacks, with attack costs diminishing as additional agents participate.*

## 6 EXPERIMENTS

To evaluate the performance of the attacks, we first measure the joint utility of the agents on the three given tasks. We use difference in utility as the primary measure of misalignment success, since misalignment impacts the ability of the agent to contribute to the joint goal. We measure privacy by comparing the actual private information to the retrieved information with LLM-as-a-judge. We assign a accuracy score of 100% if the LLM states the retrieved information is fully accurate, 50% if it is partially accurate, and 0% if it is inaccurate. We report the context overflow attack success rate (ASR) by measuring the number of times, out of 30 seeded runs, the attack causes an API error.

### 6.1 DRIVING USE CASES

***Meeting Scheduling.*** Meeting Scheduling is a partially-generated real-world environment where agents are attendees; decision variables are meetings $\mathcal{X} = \{x_m\}$ with discrete slots $\mathcal{D}_{x_m} = \{0, \ldots, 9\}$ (1-hour blocks). Each meeting has an *owner* (one of its attendees). The context $c$ provides a per-meeting mode {PHYSICAL, ZOOM} and building coordinates for physical meetings. For each agent $i$, $I_i$ specifies (a) preferred slots $P_{i,m} \subseteq \{0, \ldots, 9\}$ for each meeting $m$ they attend, and (b) a strict priority $\pi_i(m) \in \mathbb{N}$ over their meetings. Optionally, $I_i$ includes a visual panel that *mirrors* the textual fields: a graph-based campus map (nodes are buildings placed at coordinates from $c$, edges labeled by integer minute distances) and a simple 10-slot strip per meeting highlighting the agent's preferred slots $P_{i,m}$; the visual adds no scoring fields beyond what $P_{i,m}$ and $c$ already convey. For the objective score, Meeting ranges from $[0, \infty]$ with a higher score being more successful.

***Personal Assistant.*** PersonalAssistant is a fully synthetic environment where each agent $i$ chooses an outfit $x_i$ from a wardrobe; an outfit is object that contains following attributes $(\text{article}, \text{color}, \text{optional image})$. The interaction graph induces color-based factors: unary PREF_COLOR/AVOID_COLOR and pairwise MATCH_COLOR/NOT_MATCH_COLOR. Only *color* affects scoring (articles/images are presentation only). $I_i$ is text specifying the agent's color likes/dislikes and neighbor cues (e.g., "match Alice," "do not match Bob") and the description of the outfits in the wardrobe. An optional *visual panel* mirrors the text description of the wardrobe: a per-agent collage of their wardrobe options. For the objective score, PersonalAssistant ranges from $[0, 1]$ with a higher score being more successful.

***Smart-Home Assistant.*** SmartHome is fully generated with real-world meter data where agents are smart-home assistants (one per home). Each home has several tasks; for each task the agent selects a start time $x_{h,j}$ from its *allowed-start* set. The context $c$ provides a time-varying sustainable capacity profile $S[t]$. When aggregate demand exceeds $S[t]$ at slot $t$, the excess is supplied by the main—unsustainable—grid. A single high-arity factor couples all homes through this main-grid draw. For each home $h$, $I_h$ enumerates its tasks with energy *consumption* (kW), *duration* (slots), and the *allowed* start times used by the scorer. An optional visual panel *mirrors* the same information: a top bar chart of $S[t]$ over the horizon and, below, per-task horizontal segments marking allowed start windows. For the objective score, SmartHome ranges from $[-\infty, 0]$ with a higher score being more successful.

### 6.2 EXPERIMENTAL SETUP

In our experiments, we use seeded, randomized configurations of each environment, varying transition dynamics, factor graph initialization, number of agents, preferences, and constraints. To exhibit complex communication, we simulate 6 agents for PersonalAssistant, 10 agents for Meeting, and 8 agents for SmartHome.

| Model | MEETING | SMARTHOME | PERSONALASSISTANT |
|---|---|---|---|
| OpenAI GPT-4.1 | $63.17 \pm 5.18$ | $-194.87 \pm 109.44$ | $0.58 \pm 0.28$ |
| OpenAI GPT-4.1-mini | $62.33 \pm 5.76$ | $-198.59 \pm 187.94$ | $0.61 \pm 0.15$ |
| OpenAI GPT-4.1-nano | $56.6 \pm 8.07$ | $-139.96 \pm 187.94$ | $0.58 \pm 0.13$ |

Table 1: Joint-utility value across different models and environements

| Category | Attack | Metric | Value | $F_{\text{POST}}$ |
|---|---|---|---|---|
| Confidentiality | Information leakage | Correctness (%) | 100% | - |
| Integrity | Adversarial agent | Utility Diff. | +0.03 | $62.30 \pm 6.17$ |
| | Com. poisoning (1-shot) | | +0.63 | $61.7 \pm 4.96$ |
| | Com. poisoning (2-shot) | | +0.73 | $61.6 \pm 5.67$ |
| | Com. poisoning (3-shot) | | +1.73 | $60.6 \pm 3.44$ |
| Availability | Context overflow | ASR (%) | 100% | - |

Table 2: Performance of Confidentiality, Integrity and Availability attacks on schedule meeting task using OpenAI GPT-4.1-mini.

## 6.3 EXPERIMENTS

Table 1 reports the joint objective value across three model and tree domains, which we use as the baseline for our attack experiments in Table 2. To study the different attacks, we adopt GPT-4.1-mini as our primary model and meeting scheduling as our primary task. We evaluate communication poisoning under varying numbers of poisoning shots to analyze the correlation between increased poisoning and misalignment. Our results show that in the privacy attack, the adversary was able to query the model and retrieve details with 100% accuracy. We also successfully conducted context overflow attacks with 100% accuracy. This shows MASs can be highly vulnerable to privacy and availability attacks.

Although we consistently observe a decrease in utility under adversarial agent and communication poisoning attacks, the reduction is not significant—indicating that while these attacks succeed, their impact remains relatively weak. However, we do observe that increasing the number of shots leads to higher attack efficacy, showing a clear correlation between poisoning percentage and attack success.

## 7 CONCLUSION

We introduce a simple, abstract framework, TERRARIUM, for studying behavior, alignment, and security of multi-agent systems (MAS) in a controlled and scalable environment. Our design is modular and extdenable and allow to study key attack vectors that MAS deployed in real-world could be exposed to: malicious agents, compromised communications, and misaligned goals.

***Limitations and Extensions.*** This framework can enable studying agents in controlled and isolated environments. It may be useful for designing and optimizing defenses and mitigations, but its simplistic design is not meant for deployment due to optimizations likely required in the communication protocol and communication proxy for memory and computational efficiency such as minimizing the memory utilization in blackboards. In future work, we plan to explore other MAS environments that involve competition and negotiation, opening up new attack vectors tied to self-interested agents and potential defense mechanisms.

## ETHICAL CONSIDERATION

TERRARIUM enables analysis of complex attack vectors on multi-agent systems, further advancing our understanding of how to deploy MAS in the real world. We do not use any real user data, nor study deployed systems, instead focusing on creating an isolated environment similar to how Wireshark (Combs, 1998) enables studying network security. We hope that our framework will

enable development of new defenses and mitigation methods under various scenarios and provide a necessary playground to isolate and inspect the attacks.

## REPRODUCIBILITY STATEMENT

We ensure reproducibility by committing to release the full TERRARIUM framework including configurations, results, and logs. Experiments were conducted with OpenAI backbones through official APIs. Detailed instructions, including environment setup and scripts to regenerate all tables/figures, will be provided in the project repository for the camera-ready version.

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

## A  DCOP GLOSSARY

We use $i \in A$ for agents, $\alpha$ for factor indices, $x \in X$ for decision variables, $S_\alpha \subseteq X$ for a factor scope, $c \in C$ for context, $z_i = o_i(c)$ for agent-$i$'s local context, $X_i = \{x \in X : \sigma(x) = i\}$ for the set of variables controlled by agent $i$, $x_{X_i} \in \prod_{x \in X_i} D_x$ for agent-$i$'s joint decision, and $x \in \prod_{x \in X} D_x$ for a global assignment.

- $A$: finite set of agents, e.g., $A = \{1, \ldots, n\}$.
- $H$: set of humans.
- Let $\eta : A \to 2^H \setminus \{\varnothing\}$ be the mapping function indicating the nonempty set of humans it serves, allowing multiple humans to share an agent and any human to control many agents. This pairing induces a many-to-many control structure that routes instructions and constraints, shaping how the underlying problem is partitioned and coordinated across agents.
- $X$: set of *decision variables* (an agent may control zero, one, or many variables).
- $D$: per-variable finite domains, $D = \{D_x \subset \mathbb{X}_x : x \in X\}$ with $x \in D_x$.
- $\sigma$: ownership map $\sigma : X \to A$; for each agent $i$, define $X_i = \{x \in X : \sigma(x) = i\}$.
- $C$: (global) *context* space; realizations denoted $c \in C$.
- $\mu$: reference distribution on $C$ used for context-averaged objectives.
- $o$: local context observation maps $o = \{o_i : C \to Z_i\}_{i \in A}$; agent $i$ observes $z_i = o_i(c)$.
- $\rho$: instruction renderer $\rho : \big(F_i^\star, c\big) \mapsto I_i$ in some modality (e.g., text/image). Agents observe $I_i$ (and possibly $z_i$), not $F_i^\star$.
- $F^\star$: ground-truth *local utilities* (asymmetric factorization). For each agent $i$ there is a finite set $F_i^\star = \{f_{i,\beta}^\star\}_{\beta=1}^{k_i}$ with scopes $S_{i,\beta} \subseteq X$ and

$$f_{i,\beta}^\star : \Big( \prod_{v \in S_{i,\beta}} D_v \Big) \times C \to \mathbb{R}, \qquad (x_{S_{i,\beta}}, c) \mapsto f_{i,\beta}^\star(x_{S_{i,\beta}}; c).$$

The union $F^\star = \bigcup_{i \in A} F_i^\star$ is the set of all factors.

- $\Pi$: policy classes, one per agent, $\Pi = \{\Pi_i\}_{i \in A}$. A (possibly randomized) policy $\pi_i \in \Pi_i$ maps local context to a *joint* distribution over the variables agent $i$ controls:

$$\pi_i : Z_i \to \Delta\Big( \prod_{x \in X_i} D_x \Big), \qquad x_{X_i} \sim \pi_i(\cdot \mid z_i), \; z_i = o_i(c).$$

Deterministic policies are the special case $\pi_i(z_i) \in \prod_{x \in X_i} D_x$.

