# OpenReview forum: "Terrarium:  Revisiting the Blackboard for Studying Multi-Agent Attacks"
_ICLR.cc/2026/Conference — ICLR 2026 Conference Withdrawn Submission_

### Official Review · Reviewer_gzrj · 2025-10-16

**Soundness:** 1
**Presentation:** 1
**Contribution:** 2
**Rating:** 2
**Confidence:** 4

**Summary:**

This paper proposes TERRARIUM, a modular and configurable framework for studying the safety, privacy, and security of LLM-powered multi-agent systems (MAS). Repurposing the classic blackboard architecture, TERRARIUM enables collaborative multi-agent tasks while supporting systematic analysis of key attack vectors, including misalignment, malicious agents, compromised communication, and data poisoning. The framework defines core abstractions (agents, environment, blackboards, tools, communication protocol) and is evaluated across three use cases (meeting scheduling, personal assistant, smart-home assistant) with four representative attacks. Experimental results show that TERRARIUM effectively measures MAS utility under normal conditions and quantifies attack impacts (e.g., 100% success rate for privacy leakage and context overflow attacks), providing a valuable testbed to advance research on trustworthy MAS.

**Strengths:**

The paper contributes by proposing TERRARIUM, a modular and configurable framework that repurposes the classic blackboard architecture in multi-agent systems (MAS) to enable fine-grained studies on the safety, privacy, and security of LLM-powered MAS—supporting multi-agent collaborative tasks, identifying key attack vectors (e.g., misalignment, malicious agents, compromised communication, poisoned data), and demonstrating its flexibility through three implemented scenarios and four representative attacks, thereby providing necessary tools to advance research toward trustworthy MAS.

**Weaknesses:**

- The paper suffers from disorganized structure and ambiguous content linkage. Core modules of the TERRARIUM framework (e.g., agents, blackboards, communication protocols) are scattered across Chapters 1, 4, and 5 without a coherent logical thread, forcing readers to cross-reference disjoint sections to grasp their interactions. Additionally, the mapping between attack vectors (e.g., confidentiality breaches) and experimental scenarios (e.g., Meeting Scheduling) is unstated, and mathematical formalizations of DCOPs lack intuitive textual explanations, creating unnecessary barriers to understanding.
- A large portion of the paper’s foundational references and core concepts are over a decade old. Key works on MAS (e.g., Wooldridge, 2009) and blackboard architectures (e.g., Erman et al., 1980b) date back to the 1980s–2000s, while recent advances in LLM-driven MAS security (post-2023) are only superficially cited without deep integration. The "revisited blackboard design"—a central idea—lacks innovative adaptations to LLM-specific challenges (e.g., long-context inefficiency), making the framework feel like a repackaging of legacy ideas rather than a forward-looking contribution.
- The paper fails to provide sufficient details for experimental reproducibility. Critical parameters are omitted, and metrics like "FPOST" in attack experiments are undefined. Claims such as "a single adversarial agent misaligns the entire system" rely on anecdotal observations rather than rigorous, replicable evidence, undermining the scientific validity of its insights and limiting their value for follow-up research.
- No tangible artifacts (e.g., open-source code, configurable prototypes, test datasets) are provided, despite the paper’s claim of a "modular framework." The promised release of the TERRARIUM framework is conditional on "camera-ready acceptance," offering no immediate utility for review. Moreover, the framework’s simplistic design (admittedly unoptimized for memory/computational efficiency) lacks engineering feasibility for real-world deployment, and no comparative analysis with existing MAS platforms (e.g., AgentDojo) validates its practical advantages.

To summarize, The paper’s central value proposition is ambiguous. It frames TERRARIUM as advancing "trustworthy MAS research" but provides neither novel theoretical breakthroughs (e.g., new attack mechanisms) nor practical tools (e.g., deployable defenses). Its modularity and attack vector analysis—touted as strengths—are either generic to MAS research or confirm known vulnerabilities, leaving the community with no distinct, actionable contributions to advance LLM-driven MAS security.

**Questions:**

Is it possible to provide any tangible artifact to support the contribution proposed in the manuscript?

---

### Official Review · Reviewer_fgDi · 2025-10-27

**Soundness:** 3
**Presentation:** 3
**Contribution:** 2
**Rating:** 4
**Confidence:** 3

**Summary:**

This paper proposes the TERRARIUM framework for fine-grained studies on safety, privacy, and security. It adapts the blackboard design—an early multi-agent systems approach—to create a modular and configurable environment that supports multi-agent collaborative tasks using large language models (LLMs), with a focus on distributed constraint optimization problems (DCOP).

**Strengths:**

Conceptual Contribution: Systematically defines the problem space by identifying and categorizing key attack vectors (misalignment, malicious agents, compromised communication, poisoned data) in LLM-powered multi-agent systems.

Infrastructural Contribution: Provides a reproducible and extensible experimental platform (TERRARIUM), enabling controlled studies on multi-agent safety, security, and privacy through a modular, blackboard-based framework.

**Weaknesses:**

Some typos like "tree" and "extdenable".

The evaluation is confined to cooperative Distributed Constraint Optimization Problems (e.g., meeting scheduling). This fails to demonstrate the framework's applicability to more complex, adversarial, and dynamic MAS settings (e.g., auctions, autonomous driving, mixed-motive games) where critical security challenges often arise. The attacks feel contrived within these simplified, structured problems. Consequently, the proposed threat taxonomy lacks validation in scenarios involving strategic deception, long-term planning, or open-ended interaction—cornerstones of real-world MAS security crises.

The experimental scope is limited by its exclusive reliance on models from the GPT family.

**Questions:**

NA

---

### Official Review · Reviewer_tzyh · 2025-10-31

**Soundness:** 2
**Presentation:** 3
**Contribution:** 2
**Rating:** 2
**Confidence:** 3

**Summary:**

This paper proposes a testing framework for evaluating the security of multi-agent LLMs, where these agents interact to collaboratively solve Distributed Constraint Optimization Problems (DCOPs). The work appears to be an engineering report describing software rather than a research paper presenting scientific findings. Therefore, I recommend submitting it to a demo track.

**Strengths:**

S1. The writing is clear and easy to follow.

S2. Proposing an open-source testing framework for multi-agent LLM security is important for helping researchers understand vulnerabilities in the interactions among LLM agents.

**Weaknesses:**

W1. It is unclear to me what is novel about the design of the proposed testing framework, TERRARIUM, compared to existing multi-agent LLM simulation frameworks. The authors list desired properties of TERRARIUM, including modularity and configurability, but these appear to be merely fundamental requirements in traditional software design.

W2. It is unclear whether the authors will release a powerful software tool for studying multi-agent LLM security. If the testing framework is described only in the paper, its contribution to advancing LLM security research will be limited, particularly given the omission of many implementation details.

W3. The experiments in this paper are highly insufficient. The authors only tested GPT-4.1 models. Furthermore, the experiments do not clarify how simulations conducted within the TERRARIUM framework accurately reflect real-world challenges.

**Questions:**

Q1. The TERRARIUM framework assumes that all agents share the same objective, which is unrealistic for cross-domain collaboration scenarios. I wonder whether TERRARIUM can be extended to accommodate LLM agents with non-aligned objectives.

---

### Official Review · Reviewer_KxoD · 2025-11-01

**Soundness:** 2
**Presentation:** 1
**Contribution:** 2
**Rating:** 4
**Confidence:** 3

**Summary:**

This paper introduces the TERRARIUM framework, designed to facilitate the study of multi-agent systems using LLMs  in a controlled environment. It leverages a blackboard architecture to create a modular and configurable playground for evaluating MAS behaviors, security vulnerabilities, and collaboration strategies. The authors explore various attack vectors such as misalignment, data stealing, and communication poisoning, and implement them across three distinct use cases (meeting scheduling, smart home assistant, and personal assistant). Through this framework, they aim to provide a testbed for researching the privacy, security, and safety of MASs in real-world environments.

**Strengths:**

1. The introduction of the TERRARIUM framework represents an innovative approach to studying multi-agent systems, especially in terms of their security, privacy, and vulnerability to adversarial attacks. This framework is highly modular, allowing for flexible experimentation with various agent types, communication protocols, and attack scenarios.

2. The paper presents a broad spectrum of attack scenarios and provides a detailed look at how each one affects the behavior and utility of MASs. The modularity of the framework enables the replication of these attack scenarios across various configurations, contributing to the study of MAS security.

**Weaknesses:**

1. I think the most important limitation is that the writing of the paper could be improved, especially in Section 3.1, where the extensive use of symbols and notations is presented. This section refers to a large number of symbols that are only explained in the appendix, and much of the content appears redundant. A more concise presentation would improve readability—this section could be reduced by more than 50% since much of the detailed notation is not crucial for the later discussions. A more intuitive or simplified approach would make the technical content more accessible to a wider audience.

2. The framework design lacks sufficient explanation regarding its background and motivation. While the blackboard architecture is referenced, there is little discussion of why this particular design was chosen or how it directly addresses the specific challenges of multi-agent system security. The authors should provide a clearer rationale for the design choices made, ideally drawing from real-world applications or theoretical foundations. The paper would benefit from a stronger connection to practical relevance and real-world scenarios to justify the chosen approach.

3. The paper presents interesting experiments, but there is a lack of quantitative analysis and ablation studies.  More in-depth quantitative analysis, such as measuring attack success rates, utility degradation, and performance under different attack scenarios, would make the results more compelling. Additionally, ablation studies could help identify the critical components of the framework and demonstrate which parts are most influential in achieving robust performance under adversarial conditions.

4. While the paper thoroughly examines various attack scenarios, it provides limited discussion of defense mechanisms against these attacks. A more balanced approach would involve not only presenting vulnerabilities but also proposing and evaluating potential mitigation strategies. The paper could explore how to design more secure MASs or discuss existing solutions for addressing these vulnerabilities, which would be particularly valuable for researchers and practitioners working on securing multi-agent systems.

**Questions:**

See Weakness.

---

### Note · Authors · 2025-11-16

I have read and agree with the venue's withdrawal policy on behalf of myself and my co-authors.